# Hexadecyltrimethylammonium hydroxide promotes electrocatalytic activity for the oxygen evolution reaction

Yugan Gao[1], Chengqi Wu[1], Sen Yang[1] & Yiwei Tan [1✉]

The oxygen evolution reaction is an essential factor in many renewable energy technologies, such as water splitting, fuel cells, and metal–air batteries. Here we show a unique solution to improve the oxygen evolution reaction rate by adjusting the electrolyte composition via the introduction of hexadecyltrimethylammonium hydroxide into an alkaline electrolyte. The strong adsorption of hexadecyltrimethylammonium cations on the surface of electrocatalysts provides the increased absolute number of $OH^-$ ions near the electrocatalyst surface, which effectively promotes the oxygen evolution reaction performance of electrocatalysts, such as $Fe_{1-y}Ni_yS_2@Fe_{1-x}Ni_xOOH$ microplatelets and $SrBaNi_2Fe_{12}O_{22}$ powders. Meanwhile, we present an electrochemical conditioning approach to engineering the electrochemically active surface area of electrocatalysts, by which the resultant $Fe_{1-y}Ni_yS_2@Fe_{1-x}Ni_xOOH$ microplatelets have a larger electrochemically active surface area after the electrochemical conditioning of the as-synthesized $Fe_{1-y}Ni_yS_2$ microplatelets using ammonia borane than those obtained after the conventional electrochemical conditioning without ammonia borane, presumably due to the appropriate conversion rate of $Fe_{1-x}Ni_xOOH$ shells.

[1] State Key Laboratory of Materials-Oriented Chemical Engineering, School of Chemistry and Chemical Engineering, Nanjing Tech University, 211816 Nanjing, China. ✉email: ytan@njtech.edu.cn

Water electrolysis to produce hydrogen and oxygen gases has been viewed as a reliable means for large-scale storage of energy obtained from intermittent sources, such as the sun, wind, and other renewable sources[1–3]. However, the electrolysis of water currently requires large energy consumption because of the slow kinetics of the oxygen evolution reaction (OER)[4,5]. The applied voltages across electrolyzer are practically in substantial excess of the thermodynamic potential (1.23 V) for water-splitting ($H_2O = H_2 + 1/2O_2$). Therefore, the design and fabrication of more advanced, inexpensive electrodes with high catalytic activity, stability, and long durability have been the pursuing goal to promote the innovation of water electrolysis devices in the past decade[6–9]. Over the past few years, many emergent and deliberate strategies, such as modification of atomically thin MXenes[10], harnessing the multiple advantages of ternary metal nanomaterials[11], modulation of electronic configuration[12,13], have been conceived as the higher levels of developing OER electrocatalysts. A series of high-performance OER electrocatalysts, such as Ni–Fe oxyhydroxides[14–17], vanadium-based oxyhydroxides/hydroxides[18], cobalt-based nanomaterials[11,19,20], metallic sulfides derived from metal organic frameworks[21,22], and perovskite oxides[23,24], have received intensive interest due to their remarkable activities, excellent stability, and low cost, which may expedite their widespread commercialization.

So far, engineering the nanoarchitecture and composition of electrocatalysts is the most effective strategy to boost the OER activity. In particular, effective synthesis approaches to electrocatalysts with a high electrochemically active surface area (ECSA) always highly desirable, but has not yet been effectively explored. However, high-ECSA electrode architectures are not always easy to achieve, for example, processing powder-based electrocatalysts, typically such as the perovskite powders, into porous and/or tiny structures with ultrathin dimensions. On the other hand, electrolyte is an alternatively important parameter to determine the water electrolysis efficiency, but has largely been ignored. Therefore, of particular interest to us is the modulation of electrolyte composition to increase the OER activity, which can make up the shortfall in ECSA of powdered electrocatalysts. Alkaline conditions can provide more favorable OER kinetics and higher stability for electrocatalysts than other electrolytes[6,25,26]. However, there has been no previous work on the effect of alkaline electrolyte composition on OER catalysis of electrocatalysts.

Herein, we firstly report an effective electrochemical conditioning (ECC) method to improve ECSA and porous features of electrocatalysts. In contrast to the conventional ECC in pure electrolyte medium, an additive, ammonia borane (AB), was introduced into the electrolyte, in order to provide a mild electrochemical roughening pretreatment. Typically, highly active Ni-doped FeOOH ($Fe_{1-x}Ni_xOOH$) nanoflakes grown on Ni-doped FeS$_2$ ($Fe_{1-y}Ni_yS_2$) microplatelets (denoted as $Fe_{1-y}Ni_yS_2@Fe_{1-x}Ni_xOOH/NF$ (nickel foam)) with an increased ECSA are prepared by the ECC in the presence of AB to enhance the OER activity. Furthermore, intriguingly, we discover that incorporation of hexadecyltrimethylammonium hydroxide (HTAH) into the alkaline electrolyte increases the current density ($j$) of the OER for $Fe_{1-y}Ni_yS_2@Fe_{1-x}Ni_xOOH$ by a factor of >4 at overpotential ($\eta$) of 320 mV, with peak activity at 0.02 M HTAH, relative to solely inorganic alkaline electrolyte. The enhancement effects of HTAH on the OER activity can be extended to other anodic electrocatalysts, such as Y-type hexaferrite powders, which can cure the adverse effects of their low ECSA. The mechanistic studies indicate that the strong adsorption of HTA$^+$ on the electrocatalysts surface increases the accumulation of OH$^-$ ions within the diffusion double layer, as evidenced by the variation in the zeta ($\zeta$) potential with HTAH concentration, which improves the OER kinetics.

## Results and discussion

**Morphology and structure of $Fe_{1-y}Ni_yS_2@Fe_{1-x}Ni_xOOH$.** The synthetic process for the $Fe_{1-y}Ni_yS_2@Fe_{1-x}Ni_xOOH$ microplatelets with hierarchical nanoarchitecture (referred to as "as-prepared") involves four sequential steps, and starts from the initial uniform-sized octahedral MIL-101 Fe precursor prepared via a hydrothermal reaction reported by Lin et al. (Supplementary Fig. 1)[27], subsequent a sulfidization process of the MIL-101 Fe under hydrothermal conditions for its conversion to iron sulfides microplatelets in a hexagonal shape (Supplementary Fig. 2 and the following Supplementary Discussion), thereafter annealing of the microplatelets supported on NF to form Ni-doped FeS$_2$ microplatelets anchored on NF ($Fe_{1-y}Ni_yS_2/NF$, see Supplementary Figs. 3 and 4, and the Supplementary Discussion behind Supplementary Fig. 4), to the ultimate ECC of the $Fe_{1-y}Ni_yS_2/NF$ in 0.1 M KOH containing 0.02 M NH$_3$BH$_3$ by cyclic voltammetry (CV) cycling. The low-magnification scanning electron microscopy (SEM) image in Fig. 1a unambiguously displays that the as-prepared product is regular hexagonal microplatelets, and well inherits the overall shape from its nearest precursor, $Fe_{1-y}Ni_yS_2$. The medium-magnification SEM images illustrate that the surface of the resulting microplatelets has been dramatically roughened compared with the $Fe_{1-y}Ni_yS_2$ microplatelets, with a relatively smooth surface and numerous small $Fe_{1-x}Ni_xOOH$ nanoflakes are vertically grown on each microplatelet, and well spaced apart from each other after the ECC (Fig. 1b). A closer observation of the nanoflakes by high-magnification SEM and transmission electron microscopy (TEM) imaging shows that most of them are fragments of hexagonal nanoplatelets, have edge lengths of 100–200 nm, and intersect with each other (Fig. 1c, d and the inset in Fig. 1d). Significantly, TEM images also reveal that all the nanoflakes have the porous features spreading across their whole surface (Fig. 1d and Supplementary Fig. 5). Such a unique 3D hierarchical configuration enables sufficient exposure of the face, edge, and corner of each nanoflake, leading to both the electrolyte supremely accessible active sites and a large ECSA of the as-prepared hierarchical $Fe_{1-y}Ni_yS_2@Fe_{1-x}Ni_xOOH$ microplatelets due to the larger puffy shell to bulk core volume ratio.

As presented in Fig. 1e, the chemical composition of a randomly selected $Fe_{1-y}Ni_yS_2@Fe_{1-x}Ni_xOOH$ microplatelet was analyzed by high-angle annular dark field (HAADF)-scanning TEM (STEM)-energy-dispersive X-ray (EDX) elemental mapping images, which confirms that the as-prepared $Fe_{1-y}Ni_yS_2@Fe_{1-x}Ni_xOOH$ microplatelets are comprised of evenly distributed Fe, Ni, S, and O elements throughout the whole microplatelet. The corresponding EDX spectrum in Fig. 2a further reveals the presence of Fe, Ni, S, and O in the as-prepared $Fe_{1-y}Ni_yS_2@Fe_{1-x}Ni_xOOH$. Its average Fe/Ni/S/O atomic ratio is 29.53:3.800:59.65:7.020 by EDX spectroscopy quantification. Quantification of the Fe, Ni, and S contents is consistently obtained to be 31.91:3.94:64.15 by inductively coupled plasma optical emission spectrometry analysis. The high-resolution TEM (HRTEM) images exhibit that the nanoflakes consist of one to several stacked layered oxyhydroxide ultrathin nanosheets with highly porous features, while the layered nanosheets weakly interact with each other to allow the intercalation of water and OH$^-$ anions between them (Fig. 1f–h and Supplementary Fig. 6)[28]. The thickness of a monolayer $Fe_{1-x}Ni_xOOH$ nanosheet is estimated to be as thin as 1.5 nm by the atomic force microscopy measurements (Supplementary Fig. 7). These architectural features of the $Fe_{1-x}Ni_xOOH$ nanosheets enable the full availability of various active sites within the nanosheets, thus

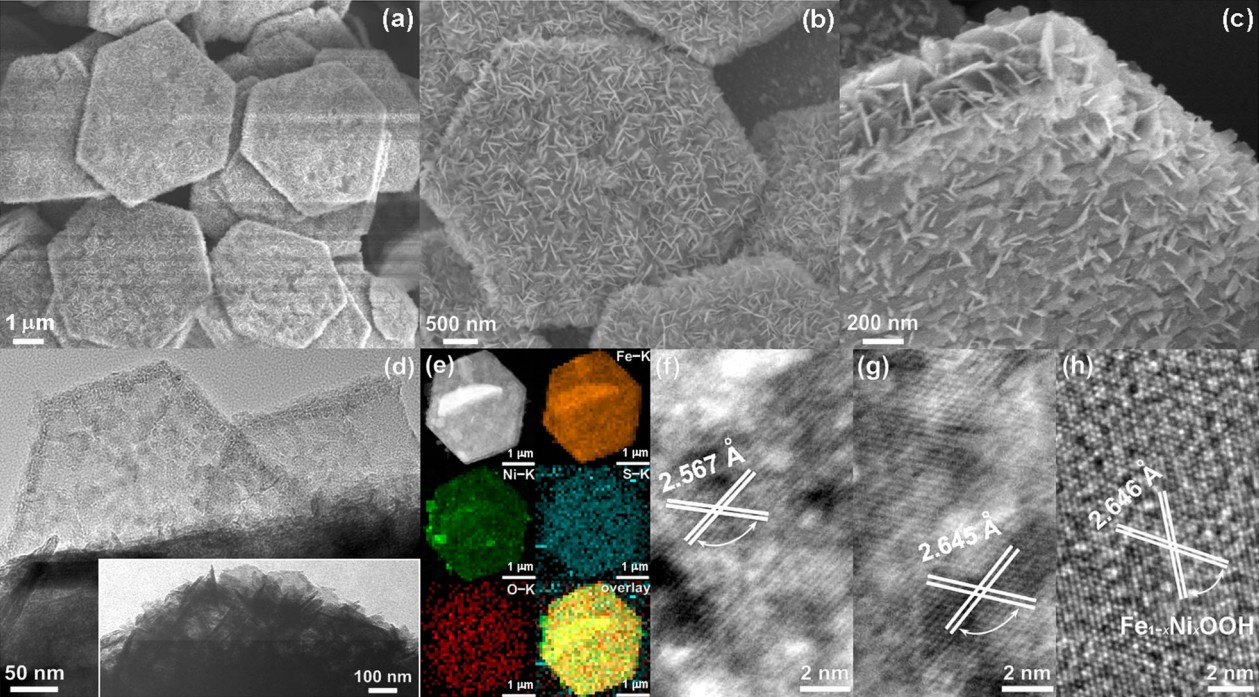

**Fig. 1 Morphological, structural, and compositional characterization of $Fe_{1-y}Ni_yS_2@Fe_{1-x}Ni_xOOH$ microplatelets. a** Low-, **b** medium-, and **c** high-magnification SEM, **d** TEM, and **e** HAADF-STEM and the corresponding HAADF-STEM-EDX elemental mapping images of $Fe_{1-y}Ni_yS_2@Fe_{1-x}Ni_xOOH$ microplatelets. **f–h** HRTEM images of the outer $Fe_{1-x}Ni_xOOH$ nanoflakes containing **f, g** several or **h** one-layered nanosheet. The inset displays the TEM image of the edge area of a randomly selected microplatelet. The magnifications for **a–c** are 6, 18, and 40 k, respectively.

further improving their advantageous effects on the ion/molecule transportation. The HRTEM images in Fig. 1f–h clearly demonstrate that the continuous, parallel lattice fringes with different interplanar spacings of 2.567 and 2.645 Å in the nanosheet domain correspond well to the lattice planes of the FeOOH phase (JCPDF no. 01-0662) and also indicate the single-crystal feature of the $Fe_{1-x}Ni_xOOH$ nanosheets. Notably, the incorporation of Ni ions into $Fe_{1-x}Ni_xOOH$ results in the increased interplanar spacings of its lattice planes, with respect to the FeOOH phase.

However, the X-ray diffraction (XRD) pattern in Fig. 2b shows that all the Bragg reflection peaks of the as-prepared $Fe_{1-y}Ni_yS_2@Fe_{1-x}Ni_xOOH$ microplatelets are perfectly attributed to the cubic pyrite-phase ($Pa3$ space group, $a = 5.428$ Å) and orthorhombic marcasite-phase ($Pmnn$ space group, $a = 4.447$ Å, $b = 5.428$ Å, and $c = 3.389$ Å) but no reflection peaks from $Fe_{1-x}Ni_xOOH$ can be detected. Therefore, it can be concluded that the bulk material (i.e., the inner core portion) is still the mixed-phase $Fe_{1-y}Ni_yS_2$, which is the predominant species in the electrocatalyst, and the surface and near-surface crystalline $Fe_{1-x}Ni_xOOH$ is the minor phase. The crystallographic structure of $Fe_{1-y}Ni_yS_2$ microplatelets is further characterized by Raman spectrum (Fig. 2c). The presence of the typical vibration features at 335 ($E_g$ vibration mode), 373 ($A_g$), and 425 cm$^{-1}$ ($T_g$) confirms the pyrite-type structure[29,30]. The strongest band at 319 cm$^{-1}$ can be attributed to $A_g$ asymmetric vibration mode of marcasite structure[29,30]. Meanwhile, a trace amount of surface oxides (most likely hematite structure) can be detected in terms of the weak bands at 221, 286, 400, 490, and 601 cm$^{-1}$ (refs. [29,31]). The Raman spectra in Fig. 2c further provide the structural difference between the as-prepared $Fe_{1-y}Ni_yS_2@Fe_{1-x}Ni_xOOH$ electrocatalyst and $Fe_{1-y}Ni_yS_2$ precursor, where the formation of the surface oxyhydroxide results in vanishing of all the peaks of FeS$_2$ phases (the probe depth of 40–50 nm for the laser source) and emerging

of new bands of the iron oxyhydroxide phase with a slightly blueshift in peak positions with respect to those of hematite phase. The high degree of porosity for the as-prepared $Fe_{1-y}Ni_yS_2@Fe_{1-x}Ni_xOOH$ microplatelets is further characterized by the N$_2$ adsorption−desorption isotherms and BJH pore-size distribution (Fig. 2d and the inset). The sample exhibits the type I isotherm curves with characteristic adsorptions portrayed by $P/P_0$ <0.1, and between 0.4 and 0.8 relating to filling of micropores (<2 nm) and mesopores (2–50 nm), respectively. Meanwhile, the capillary condensation of N$_2$ in the inter-NFs porosity and gap gives rise to the sharp uptake above $P/P_0$ of 0.8. However, the $Fe_{1-y}Ni_yS_2$ microplatelets precursor exhibits the type III isotherm curves and has the lowest porosity with only mesopores from the gaps between microplatelets. The calculated specific surface area ($S_{BET}$) of the $Fe_{1-y}Ni_yS_2@Fe_{1-x}Ni_xOOH$ microplatelets is 37.4 m$^2$ g$^{-1}$, which is much larger than that of the $Fe_{1-y}Ni_yS_2$ microplatelets precursor (12.5 m$^2$ g$^{-1}$), enabling a sufficiently high ECSA for the electrocatalyst. The surface chemical compositions and electronic states of $Fe_{1-y}Ni_yS_2@Fe_{1-x}Ni_xOOH$ microplatelets, and $Fe_{1-y}Ni_yS_2$ are further analyzed using X-ray photoelectron spectroscopy (XPS; see Supplementary Fig. 8 and the following Supplementary Discussion). The $Fe_{1-y}Ni_yS_2@Fe_{1-x}Ni_xOOH$ microplatelets have a surface Fe/Ni/S/O atomic ratio of 28.51:2.16:4.44:64.89, and show the overall increased oxidation states of Fe and Ni atoms compared with the $Fe_{1-y}Ni_yS_2$ microplatelets, as determined by the XPS analyses.

To highlight the function of AB during the ECC, a control experiment was performed, in which the pristine $Fe_{1-y}Ni_yS_2$/NF was electrochemically conditioned without AB (designated as $Fe_{1-y}Ni_yS_2$-ECC thereafter). As reported previously, metallic oxyhydroxides (in our case, $Fe_{1-x}Ni_xOOH$) are supposed to be generated after this ECC[20,32]. In contrast, after this conventional ECC, the $Fe_{1-x}Ni_xOOH$ nanoflakes formed on $Fe_{1-y}Ni_yS_2$-ECC microplatelets are far less than those on

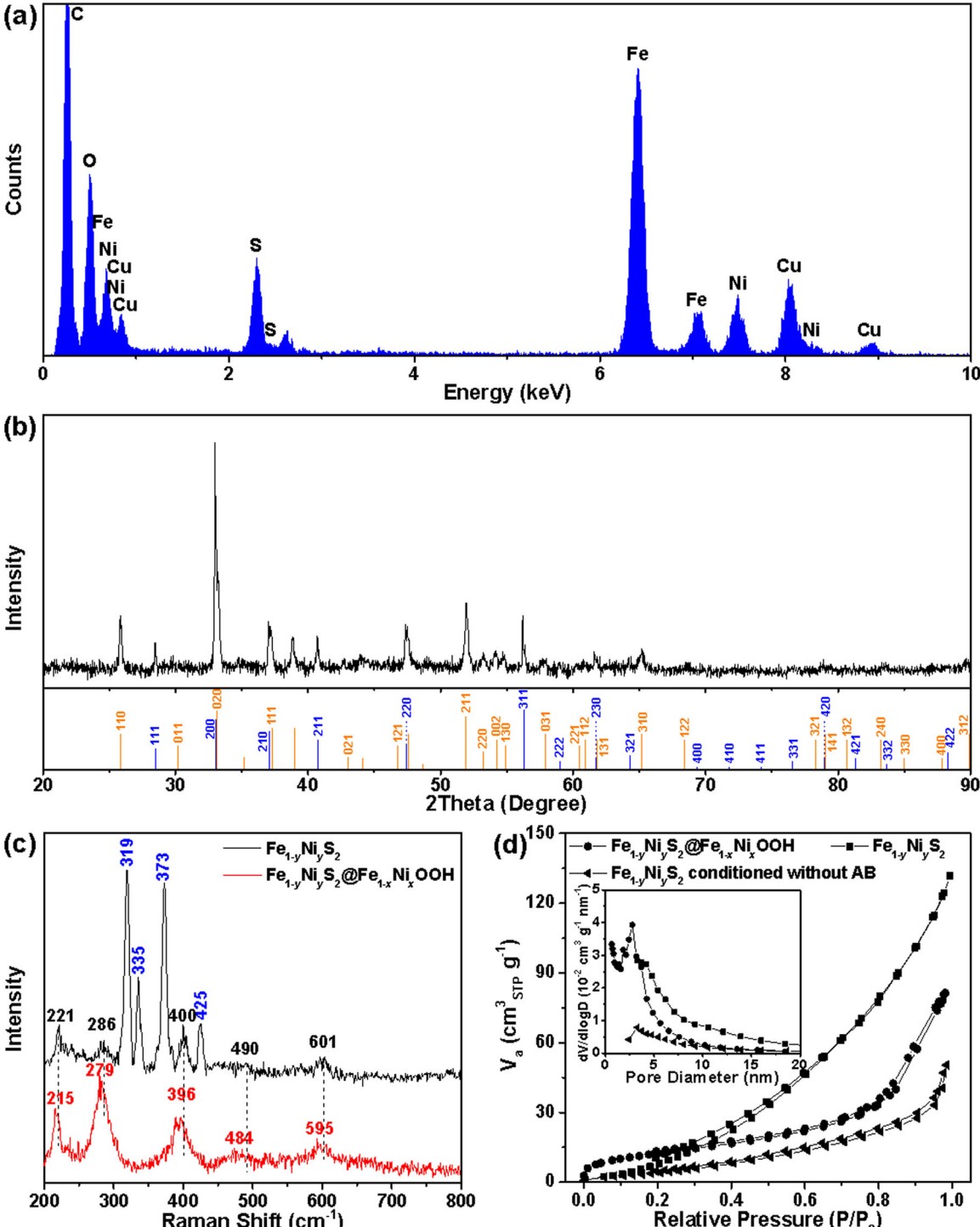

**Fig. 2 Compositional and structural characterization of $Fe_{1-y}Ni_yS_2@Fe_{1-x}Ni_xOOH$ microplatelets and comparison with the precursor and control sample. a** EDX spectrum and **b** powder XRD diffractogram of $Fe_{1-y}Ni_yS_2@Fe_{1-x}Ni_xOOH$ microplatelets. The Cu and C signals in **a** originate from the carbon-coated copper grid used for TEM imaging. The intensities and positions for the pure pyrite (blue, JCPDF no. 26-0801) and marcasite (orange, JCPDF no. 02-0908) references are given as different colorful bars at the bottom of **b** according to the JCPDS database. Comparison of **c** the Raman spectra, and **d** $N_2$ adsorption−desorption isotherms and BJH pore-size distribution plots (the inset) of the $Fe_{1-y}Ni_yS_2@Fe_{1-x}Ni_xOOH$, $Fe_{1-y}Ni_yS_2$ electrochemically conditioned without AB, and $Fe_{1-y}Ni_yS_2$ microplatelets.

$Fe_{1-y}Ni_yS_2@Fe_{1-x}Ni_xOOH$ microplatelets (Supplementary Fig. 9). The calculated $S_{BET}$ of the $Fe_{1-y}Ni_yS_2$-ECC is 18.2 m$^2$ g$^{-1}$ and much lower than that of the $Fe_{1-y}Ni_yS_2@Fe_{1-x}Ni_xOOH$ due to its low porosity with an overwhelming majority of mesopores from the gaps between microplatelets (Fig. 2d and the inset in it). Evidently, AB plays an important role in tuning the surface

morphology during the ECC. AB is a strong reducing agent and can be electro-oxidized during the CV cycling, as shown in Supplementary Fig. 10. The electron exchange between the $Fe_{1-y}Ni_yS_2$ microplatelet and AB decelerates the conversion of $Fe_{1-y}Ni_yS_2$ to oxidized oxyhydroxide, thereby allowing the oxyhydroxide to grow to its full potential. In addition, the lattice

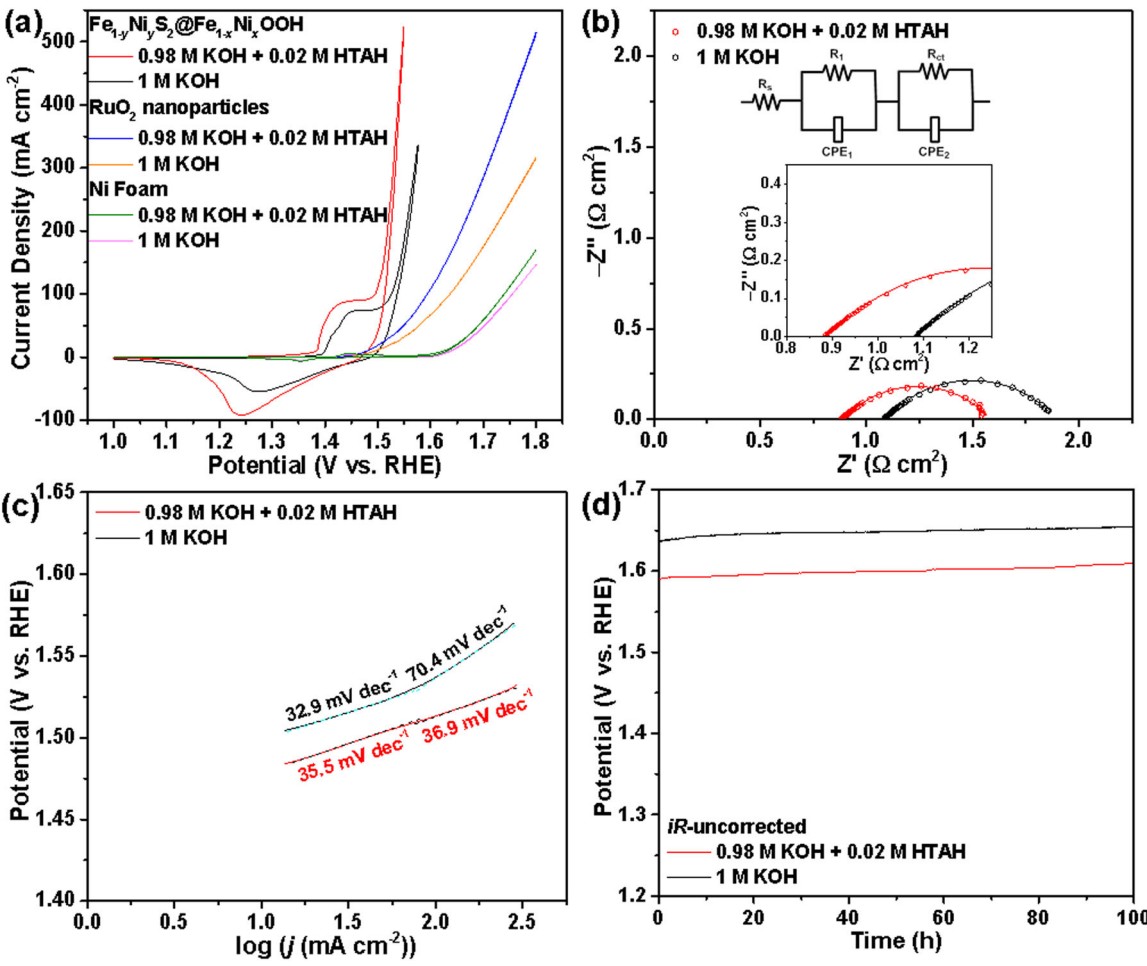

**Fig. 3 Comparison of the OER electrocatalytic performance of $Fe_{1-y}Ni_yS_2@Fe_{1-x}Ni_xOOH/NF$ electrodes in 1 M KOH and 0.98 M KOH + 0.02 M HTAH.**
**a** Comparison of *iR*-corrected CV curves recorded at a scan rate of 5 mV s$^{-1}$. **b** EIS Nyquist plots. **c** *iR*-corrected polarization curve-derived Tafel slopes.
**d** *iR*-uncorrected CP curves recorded at a constant *j* of 150 mA cm$^{-2}$. Insets: the fitted EEC used to model the electrode systems (top) and Nyquist plots on a smaller scale (bottom).

strain induced by the M(II)/M(III) conversion during ECC is believed to be the major contributor to the generation of numerous nanopores. On the other hand, AB molecules and their oxidized intermediates may serve as the effective capping agents to control the morphology and thickness of the oxyhydroxide nanosheets, leading to the formation of the expanded and ultrathin nanosheets.

**Effects of HTAH on the OER performance.** The electrocatalytic activities of various electrodes toward the OER were evaluated in a conventional three-electrode cell by CV in 1 M OH$^-$ ions consisting of only KOH or KOH (0.98 M) and HTAH (0.02 M; denoted as KOH + HTAH thereafter). The *iR* (*i*, current; *R*, series resistance derived from the impedance measurements)-corrected polarization curves in Fig. 3a and Supplementary Fig. 11a illustrate that the $Fe_{1-y}Ni_yS_2@Fe_{1-x}Ni_xOOH/NF$ exhibits a much better catalytic activity than the $Fe_{1-y}Ni_yS_2$-ECC/NF, $Fe_{1-y}Ni_yS_2$/NF, and RuO$_2$/NF benchmark in both electrolytes because the OER on the former requires a lower $\eta$ than the later three ones at the same *j*, especially at a large *j*. For example, to reach *j* of 50 mA cm$^{-2}$ during the negative-going scan, which was adopted to eliminate the interference of oxidation waves from Ni species, the $Fe_{1-y}Ni_yS_2@Fe_{1-x}Ni_xOOH/NF$ requires an $\eta$ of 273 and 292 mV in KOH + HTAH and KOH alone, respectively (note that all the following values in different electrolytes will be given

in the same order as above), which is much smaller than those of the $Fe_{1-y}Ni_yS_2$-ECC/NF (333 and 350 mV), $Fe_{1-y}Ni_yS_2$/NF (361 and 385 mV), and RuO$_2$/NF (319 and 348 mV). Furthermore, the catalytic *j* predominantly originates from the active materials rather than the NF substrate, since the *j* of NF only accounts for 1–5% of the overall *j* of each electrode in the $\eta$ range of 270–370 mV. More importantly, all the samples are apparently endowed with a higher activity toward the OER in KOH + HTAH than in KOH alone as the *j* values of the $Fe_{1-y}Ni_yS_2@Fe_{1-x}Ni_xOOH$, $Fe_{1-y}Ni_yS_2$-ECC/NF, $Fe_{1-y}Ni_yS_2$/NF, and RuO$_2$ increase by 313%, 92.5%, 418%, and 60.3%, respectively, based on the average of the *j* during the positive- and negative-going scans at $\eta = 320$ mV in the presence of a small amount of HTAH (0.02 M).

The vastly superior OER performance of the $Fe_{1-y}Ni_yS_2@Fe_{1-x}Ni_xOOH/NF$ in KOH + HTAH originates from fast charge-transfer kinetics, as evidenced by the electrochemical impedance spectra (EIS) in Fig. 3b and Supplementary Fig. 11b, d. The Nyquist plots (symbols) of EIS centered at 1.57 V versus reversible hydrogen electrode ($V_{RHE}$) according to modeling with the corresponding electric equivalent circuit (EEC) reveal the presence of two overlapped semicircles at high frequencies and low frequencies, respectively (insets in Fig. 3b and Supplementary Fig. 11b, d. Also see the Supplementary Discussion behind Supplementary Fig. 11). The charge transfer resistance ($R_{ct}$) value of each sample in KOH + HTAH is smaller than that in KOH

alone, suggesting the superior interfacial charge-transfer kinetics in the mixed electrolyte (Supplementary Table 1). Meanwhile, among all the samples, the smallest $R_{ct}$ values obtained for the $Fe_{1-y}Ni_yS_2@Fe_{1-x}Ni_xOOH/NF$ explicitly reveal the highest charge-transfer kinetics on it. The additional discussions are provided in the Supplementary Information. To gain further insight into the OER kinetics of these samples, the Tafel slopes that have been $iR$-corrected, and obtained based on the negative-going scan curves are compared in Fig. 3c and Supplementary Figs. 11c, e. In particular, the $Fe_{1-y}Ni_yS_2@Fe_{1-x}Ni_xOOH/NF$ exhibits the lowest Tafel slopes with a value of 35.5 mV dec$^{-1}$ in the low $j$ region and 36.9 mV dec$^{-1}$ in the large $j$ region in KOH + HTAH, which is slightly higher than that (32.9 mV dec$^{-1}$) in the lower $j$ region, but much lower than the value (70.4 mV dec$^{-1}$) of the larger $j$ in KOH, respectively. These data unequivocally demonstrate that 0.02 M HTAH effectively facilitates and expedites the OER kinetics in the higher $j$. Similarly, there are the same effects of HTAH on the OER kinetics for the $Fe_{1-y}Ni_yS_2$-ECC/NF, $Fe_{1-y}Ni_yS_2$/NF, and $RuO_2$ (Supplementary Fig. 11c, e). Concomitantly, for our synthesized new samples, the Tafel slope follows the order of $Fe_{1-y}Ni_yS_2@Fe_{1-x}Ni_xOOH < Fe_{1-y}Ni_yS_2$-ECC $< Fe_{1-y}Ni_yS_2$ in both electrolytes by comprehensively considering the slope values in different $j$ regions.

The ECSAs of the $Fe_{1-y}Ni_yS_2@Fe_{1-x}Ni_xOOH$, $Fe_{1-y}Ni_yS_2$-ECC, and $Fe_{1-y}Ni_yS_2$ are evaluated in both electrolytes using the conventional electrochemical double layer capacitance ($C_{dl}$) that is determined by CV measurements within the potential window of 1.0–1.1 $V_{RHE}$ (Supplementary Fig. 12 and 13, and Supplementary Table 1). The ECSAs for each active material are very similar in both electrolytes and almost irrelevant to HTAH based on their $C_{dl}$ values (Supplementary Table 1). In particular, the ECSA of the $Fe_{1-y}Ni_yS_2@Fe_{1-x}Ni_xOOH$ is 2.3 and 3.6 times higher than that of the $Fe_{1-y}Ni_yS_2$-ECC and $Fe_{1-y}Ni_yS_2$, respectively. Note that the trend in ECSA for these samples is in good accordance with that in $S_{BET}$ described above. Apparently, the enlarged ECSA for the $Fe_{1-y}Ni_yS_2@Fe_{1-x}Ni_xOOH$ enables the exposure of more active sites on it for the OER. As demonstrated in both our experiments and others' report, the surface components of various electrocatalysts have been converted to oxyhydroxide species after the anodic ECC[20,32,33]. In our case, $Fe_{1-x}Ni_xOOH$ nanoflakes have been formed on the surfaces of all the measured electrocatalysts. Concurrently, the $FeS_2$ bulk of each active material remains unchanged due to the protective effects of compact outer $Fe_{1-x}Ni_xOOH$ shell (vide infra). Therefore, the enhanced electrocatalytic activities of $Fe_{1-x}Ni_yS_2@Fe_{1-x}Ni_xOOH$ could be definitely attributed to its dramatically enlarged ECSA, as its surface forms much more nanoflakes with fluffy and rich porous features after the unique ECC. Typically, the stability tests for the most active $Fe_{1-y}Ni_yS_2@Fe_{1-x}Ni_xOOH$ both in KOH + HTAH and KOH were evaluated separately by continuous chronopotentiometric (CP) measurement for 100 h (Fig. 3d). This electrocatalyst presents an outstanding level of stability with only a 3% or 4% increase in $\eta$ in KOH + HTAH or KOH, respectively, during 100 h continuous galvanostatic electrolysis at 150 mA cm$^{-2}$. Moreover, SEM, XRD, and HRTEM data show that there are no visible morphological and structural variations in the $Fe_{1-y}Ni_yS_2@Fe_{1-x}Ni_xOOH$ electrocatalyst in KOH + HTAH after the CP tests due to the highly stable $Fe_{1-x}Ni_xOOH$ shell in alkaline medium and its mechanically robust structure, strong binding to $Fe_{1-y}Ni_yS_2$, and compact underlying layer to protect the $Fe_{1-y}Ni_yS_2$ core (Supplementary Fig. 14). The same results are obtained for this electrocatalyst in KOH alone, but the data are not shown for brevity. The durable performance demonstrates that not only the electrocatalyst structures, but also the electrolyte containing HTAH is durable and viable for OER.

The mixed-phase $FeS_2$ with both marcasite and pyrite has been proven to have a high conductivity, independent of the content of each phase[30]. Therefore, the high conductivity of the matrix in combination with the redox hopping-type conduction achieved by the mixed Ni(II)/Ni(III) valence via Ni doping provides a high conductivity for the electrocatalyst[34,35], therefore further promoting the OER kinetics on $Fe_{1-y}Ni_yS_2@Fe_{1-x}Ni_xOOH$.

Notably, the polarization curves reveal that the $j$ of each electrode in KOH + HTAH is significantly larger than that in KOH alone at the same applied potential. As discussed above, ECSA (i.e., active site density) of each electrode, evaluated by the same $C_{dl}$ measurements in KOH + HTAH and KOH alone, however, is not perceptibly influenced by HTAH. We therefore postulate that the remarkable OER activity enhancement in KOH + HTAH could be ascribed to the modification of electrode/electrolyte interface nature and/or active site nature due to introducing HTAH. Recently, Yang et al. demonstrated that the large organic tetra-alkylammonium cations (TAA$^+$) strongly interact with the oxygen species formed upon deprotonation of Ni(Fe)OOH electrocatalysts[36]. Conversely, the catalytic OER activity, nonetheless, is degenerated upon introducing these TAA$^+$ into electrolyte in their report. We note that the Grimaud group used a higher concentration of TAA$^+$ (0.1 M) to probe the active oxygen intermediate, which leads to the negative effects on the OER activity induced by the proposed mechanism of disturbing the hydrogen bonds network at the interfacial water[36]. However, the positive function on the OER activity for TAA$^+$ that is double-edged might be disregarded due to the lack of adjusting the concentration of TAA$^+$.

Accordingly, we further studied the HTAH concentration ($C_{HTAH}$) dependence for the OER activity of the $Fe_{1-y}Ni_yS_2@Fe_{1-x}Ni_xOOH$ electrocatalyst (Fig. 4). The OER peak activity can be seen at the $C_{HTAH}$ of 0.02 M, while continuing to increase $C_{HTAH}$ to 0.04 M leads to a continual decrease in the OER activity according to the highest $j$ achieved for 0.02 M HTAH at the same $\eta$ (e.g., 323, 302, 289, 307, and 343 mV to achieve 200 mA cm$^{-2}$ for the electrolytes containing no, 0.01, 0.02, 0.03, and 0.04 M HTAH, respectively), as well as the lowest Tafel slope (70.4, 59.3, 36.9, 66.8, and 107.6 mV dec$^{-1}$ corresponding to the above $C_{HTAH}$) in the higher $j$ region (Fig. 4a, b). To further evaluate the intrinsic activity of electrocatalyst associated with HTAH, we have calculated the turnover frequency (TOF) in terms of the $Fe_{1-x}Ni_xOOH$ mass and composition (see "Methods" section) by assuming a $z = 4$ electron transfer for the water oxidation reaction using the following equation[37]:

$$TOF = \frac{i \cdot FE}{z \cdot F \cdot n_{Fe+Ni}}, \qquad (1)$$

where faradaic efficiency (FE) is determined to be 95.3% by gas chromatography (GC). Figure 4c shows the TOF values at a different $C_{HTAH}$, which are obtained based on the total number of Fe + Ni atoms from the steady-state current at an $iR$-corrected $\eta = 300$ and 320 mV. Among various electrolytes, the electrocatalyst has the highest intrinsic activity in KOH + HTAH with TOF peaking at $0.285 \pm 0.0276$ and $0.452 \pm 0.0657$ s$^{-1}$ at $\eta = 300$ and 320 mV, respectively. Note that in the range of our studies, the addition of a different HTAH concentration into KOH to maintain 1 M OH$^-$ ions gives a negligible difference in pH values ($13.64 \pm 0.10$, $13.57 \pm 0.08$, $13.69 \pm 0.09$, $13.51 \pm 0.12$, and $13.61 \pm 0.05$ for the electrolytes containing no, 0.01, 0.02, 0.03, and 0.04 M HTAH, respectively). Evidently, in our case, there should be a distinctive mechanism at play for promoting the OER activity by HTAH. Our results unequivocally suggest the positive synergistic effect between the electrocatalyst and low levels of HTAH for OER catalysis. Given that HTAH is a typical surfactant, the catalytically non-active HTA$^+$ cations in the electrolyte may impact on the activity of

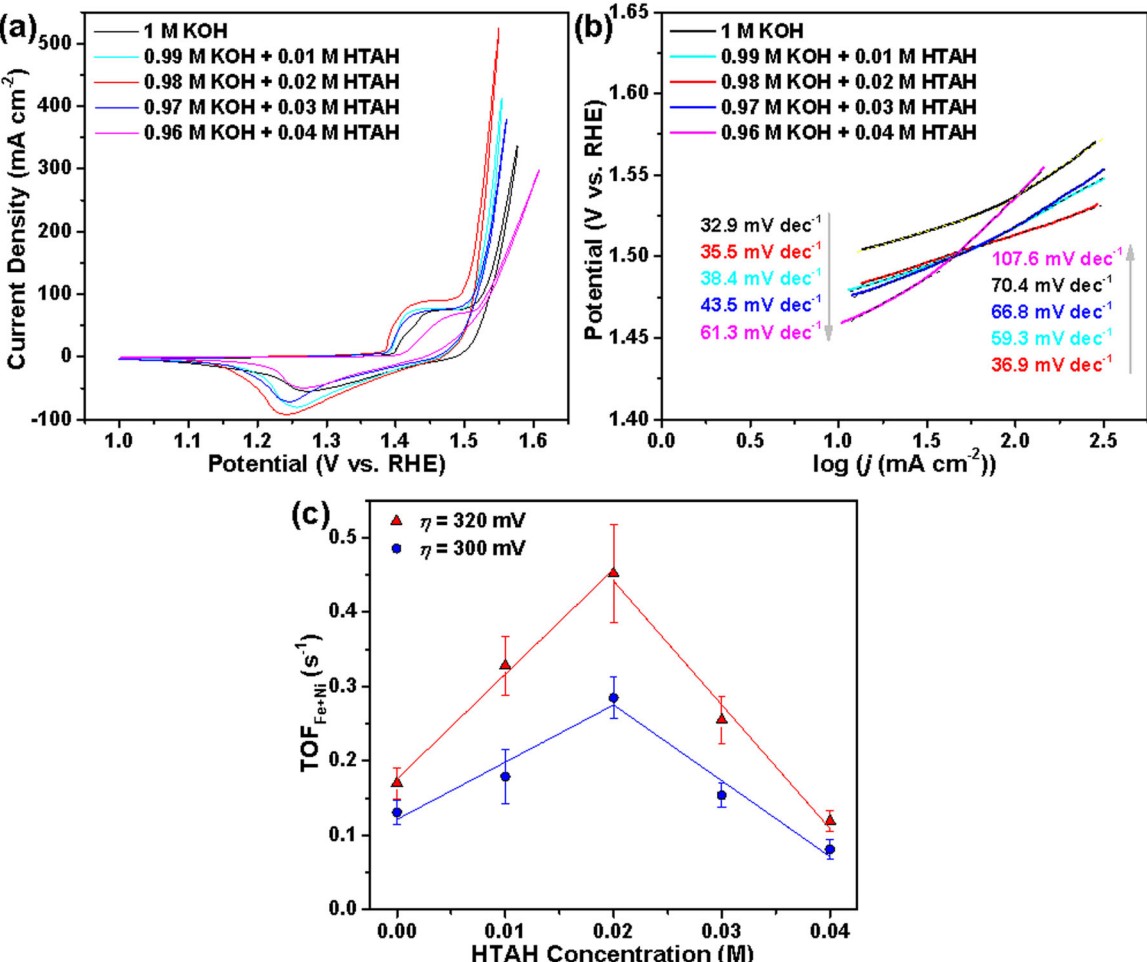

**Fig. 4 OER electrocatalytic performance of Fe$_{1-y}$Ni$_y$S$_2$@Fe$_{1-x}$Ni$_x$OOH/NF as a function of HTAH concentration. a** *iR*-corrected CV curves, **b** Tafel slopes derived from the CV curves in **a**, and **c** TOF values and the corresponding error bars (at *η* of 300 mV) for Fe$_{1-y}$Ni$_y$S$_2$@Fe$_{1-x}$Ni$_x$OOH/NF recorded in purified electrolytes containing a different concentration of HTAH. The error bars are derived from the standard deviation from independent measurements of three identically prepared samples used to obtain the average TOF.

electrocatalysts by a unique way through its strong adsorption onto the surface of electrocatalysts.

To clarify the role of HTAH in substantially enhancing the intrinsic OER activities of these electrodes, its effects are evaluated by determining the streaming potential, as a function of its concentration in electrolyte. Consequently, the zeta (ζ) potentials used to characterize the electrical properties of the Fe$_{1-y}$Ni$_y$S$_2$@Fe$_{1-x}$Ni$_x$OOH/NF electrode surface in various electrolytes containing a different concentration of HTAH were determined, and calculated from the streaming potential using the Helmholtz–Smoluchowski equation[38]:

$$\frac{dE_z}{d\Delta P} = \frac{\varepsilon_0 \varepsilon_r \zeta}{\mu \Lambda_0}, \tag{2}$$

where $E_z$ and $\Delta P$ are the measured streaming potential and applied hydraulic pressure, respectively. $\varepsilon_0$ is the permittivity of free space, $\varepsilon_r$ is the dielectric constant of the solution (59), $\mu$ is the solution viscosity (1.128 mPa s) at 25 °C, and $\Lambda_0$ is the solution conductivity (0.22 S cm$^{-1}$). All these parameters are assumed to be unchanged as there were little differences in the electrolyte concentration. Figure 5a plots the zeta potential of the surface of shear for Fe$_{1-y}$Ni$_y$S$_2$@Fe$_{1-x}$Ni$_x$OOH/NF electrode as a function of the HTAH concentration. Note that the magnitude of the zeta potential initially decreases from −98 to −136 mV with increasing the HTAH concentration from 0 to 0.01 M, and the

0.02 M HTAH electrolyte system presents the most negative electrode surface of shear with a zeta potential as low as −167 mV. The cationic surfactant is expected to impart a positive charge to the electrode surface, thereby leading to enhanced electrostatic attraction between the electrode surface/surfactant headgroup and OH$^-$ ions in electrolyte. Therefore, when the HTAH concentration is low, the strong adsorption of HTA$^+$ causes an accumulation of counterions (OH$^-$) in the diffusion double layer due to electrostatic attraction, which remarkably increases the concentration of OH$^-$ ions in the inner Helmholtz plane within the Stern layer (Fig. 5b, c). As a result, the OER rate is significantly enhanced since the absolute number of reactant OH$^-$ ions near the electrode surface increases and therefore the O*(active site)···OH$^-$ interaction probability is boosted to promote the formation of O–O bond. Note that the enlarged peak areas of the Ni$^{2+}$/Ni$^{3+}$ redox couple in Figs. 3a and 4a, and Supplementary Fig. 11a originates from the accelerated reaction rate, further indicating the increased concentration of OH$^-$ ions near the electrocatalyst surface based on the following equation:

$$\text{Ni}^{2+} + \text{H}_2\text{O} + \text{OH}^- = \text{NiOOH} + \text{e}^- + 2\text{H}^+. \tag{3}$$

However, with further increasing the concentration of HTAH >0.02 M, more HTA$^+$ is adsorbed onto the electrode surface, and therefore the surface of shear become more positive with a value

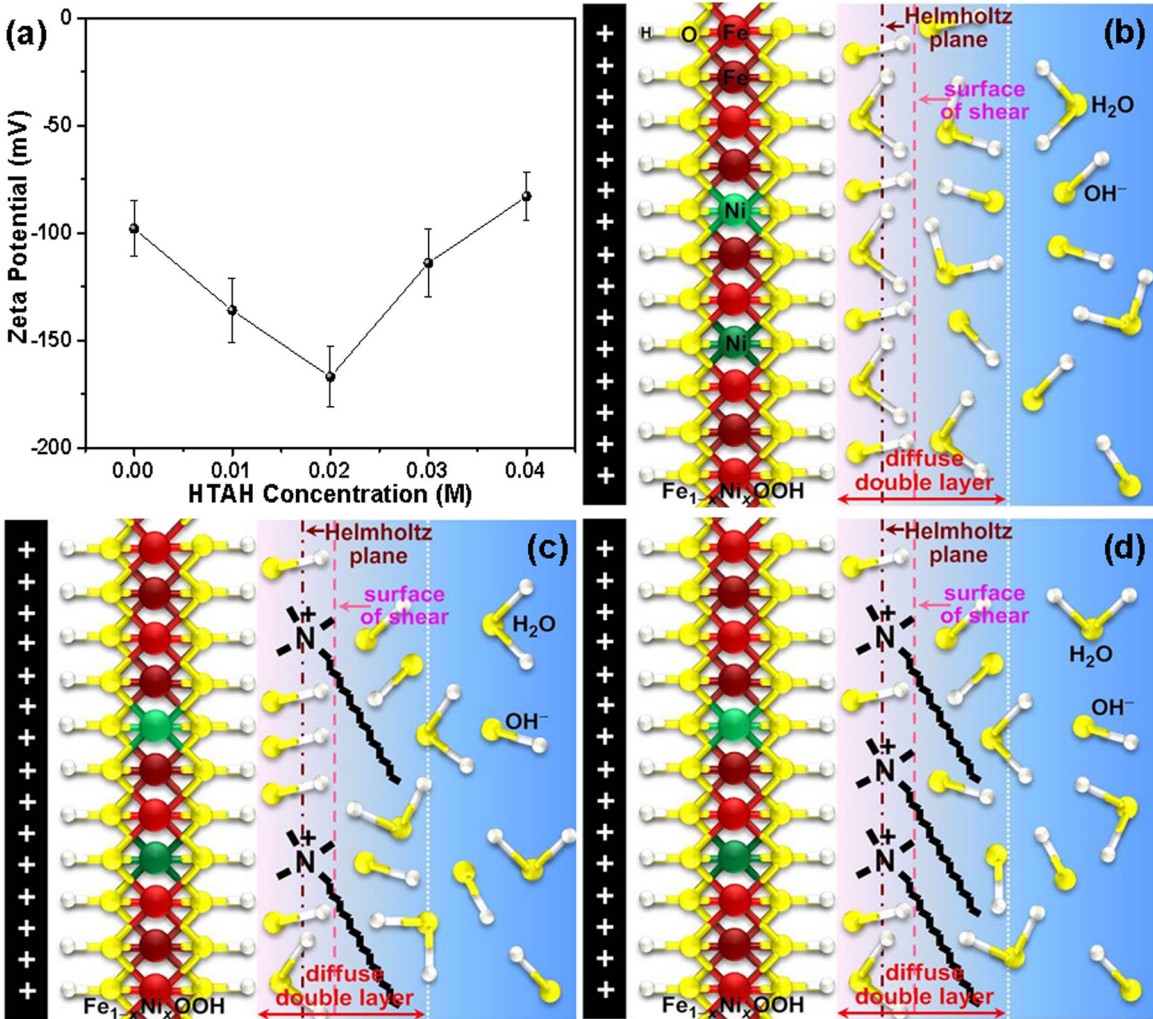

**Fig. 5 Zeta potential for different electrolytes and schematic illustrations of the interaction between HTAH and Fe$_{1-x}$Ni$_x$OOH surface. a** The zeta potential of Fe$_{1-y}$Ni$_y$S$_2$@Fe$_{1-x}$Ni$_x$OOH/NF electrode versus the HTAH concentration. The error bars are based on measurements of three identically prepared samples in triplicate. **b–d** Schematic depiction of the OH$^-$ ion distribution near this electrode surface **b** without HTAH, and with HTAH **c** at low (0.02 M) and **d** high (0.04 M) concentrations. For brevity and clarity, the couterions (K$^+$ ions) and hydration effects are not shown.

of −114 mV at 0.03 M until a value of −83 mV is reached at 0.04 M HTAH concentration. Thus, excess adsorbed HTA$^+$ cations impede the approaching of OH$^-$ ions to the active sites on electrode surface, leading to the decrease in the OER activity (Fig. 5d).

To verify the generalizability of this electrolyte strategy, we have also studied the OER activity electrocatalyzed by a Y-type hexaferrite, SrBaNi$_2$Fe$_{12}$O$_{22}$, in KOH + HTAH and KOH alone. The SEM image in Fig. 6a shows the irregular morphology and large size (1–6 μm) of the as-prepared SrBaNi$_2$Fe$_{12}$O$_{22}$ particles. The crystal structure of the SrBaNi$_2$Fe$_{12}$O$_{22}$ particles have been verified using a combination of powder XRD and HRTEM. All the Bragg reflection peaks in Fig. 6b can be perfectly indexed to a pure rhombohedral structure with the lattice parameters of $a = b = 5.844$ Å and $c = 43.33$ Å (space group $R3m$ (166), JCPDF no. 54-1165). The corresponding structure of the unit cell, which is comprised of three T and S layers are alternately stacked[39], is depicted in Fig. 6c. Moreover, the clearly resolved lattice fringes in the HRTEM image of SrBaNi$_2$Fe$_{12}$O$_{22}$ powders with interplanar distances of 2.928 Å can be definitely assigned to the (110) planes of the SrBaNi$_2$Fe$_{12}$O$_{22}$, further revealing the structural phase of Y-type hexaferrite (Fig. 6d). The OER activity of SrBaNi$_2$Fe$_{12}$O$_{22}$ was evaluated in 1.0 M KOH and

KOH + HTAH aqueous solutions. Figure 6e presents the $iR$-corrected CV curves performed at a sweep rate of 5 mV s$^{-1}$. Apparently, the hexaferrite electrocatalyst exhibits significantly improved $j$ and much decreased onset potential (1.329 versus 1.342 $V_{RHE}$) in KOH + HTAH versus KOH, indicating the OER catalytic performance of SrBaNi$_2$Fe$_{12}$O$_{22}$ is substantially better than in KOH alone. Typically, compared to KOH alone, a 7.5-fold increase in the $j$ value is achieved for the KOH + HTAH at $\eta$ of 270 mV. Meanwhile, the derived Tafel slopes are lower for KOH + HTAH than those for KOH alone both in the low and high $j$ regions (Fig. 6f). EIS measurements further validate the favorable intrinsic interfacial charge-transfer kinetics in KOH + HTAH for the OER electrocatalysis, as shown in Fig. 6g. The fitted $R_{CT}$ value for KOH + HTAH is 7.2 Ω, which is much smaller than that (11.3 Ω) for KOH alone. In addition, the SrBaNi$_2$Fe$_{12}$O$_{22}$ exhibits excellent stability both in KOH + HTAH and KOH alone, maintaining a nearly constant $\eta$ (282 and 355 mV for the KOH + HTAH and KOH, respectively) to afford a constant $j$ of 150 mA cm$^{-2}$ over 100 h, which is required for practical application in electrolyzers. To our knowledge, the currently reported electrocatalysts coupled with the new electrolyte deliver the excellent OER performance, outperforming most of bimetallic Ni−Fe-based OER electrocatalysts, such as NiFeS[40],

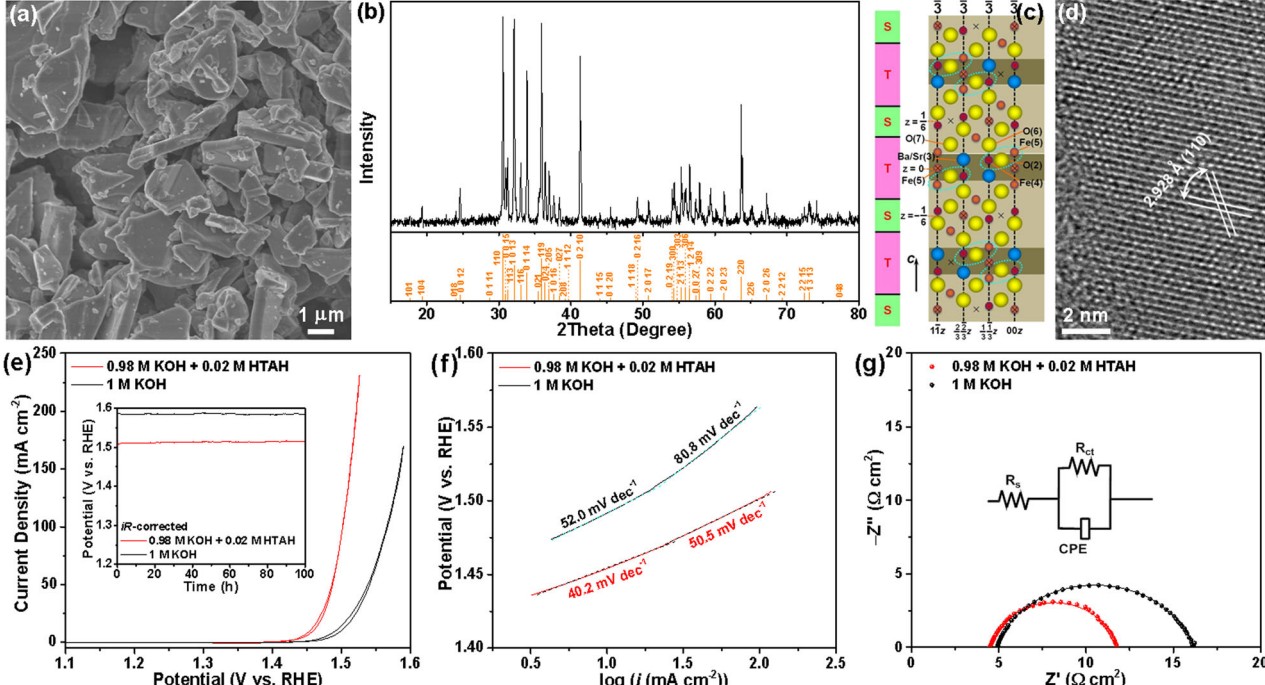

**Fig. 6 Structural characterization and OER electrocatalytic performance of Y-type hexaferrite. a** SEM image, **b** XRD pattern, **c** unit cell structure, **d** HRTEM image of the as-prepared SrBaNi$_2$Fe$_{12}$O$_{22}$, **e** iR-corrected CV curves, **f** EIS Nyquist plots centered at 1.45 $V_{RHE}$, and **g** Tafel slopes recorded the SrBaNi$_2$Fe$_{12}$O$_{22}$ supported on GCD electrode in 1 M KOH and 0.98 M KOH + 0.02 M HTAH. Insets: **e** CP curves recorded at constant $j$ of 150 mA cm$^{-2}$ and **f** Randles circuit used to fit the EIS responses, where $R_S$, $R_{CT}$, and CPE represent a series resistance, a charge transfer resistance, and a constant phase element, respectively.

Ni–Fe–OH@Ni$_3$S$_2$/NF[41], NiFe LDH@NiCoP/NF[42], NiFe (oxy) sulfide[43], Ni–Fe disulfide@oxyhydroxide[44], NiCo$_2$S$_4$ NW/NF[45], Ni$_x$Fe$_{1-x}$Se$_2$-DO[46], Ni–Fe–Se nanocages[47], amorphous NiFe[48], mesoporous Ni–Fe–O nanowires[49], Ni$_5$P$_4$/NiP$_2$/NiFe LDH[50], and Fe@NiFe LDH (Supplementary Table 2)[51], while our intention is not exclusively aimed at elaborate design the electrocatalyst composition and nanostructures. For clear comparison, the polarization curves and derived Tafel slopes of Fe$_{1-y}$Ni$_y$S$_2$@Fe$_{1-x}$Ni$_x$OOH/NF without compensation for iR drop are provided in Supplementary Fig. 15.

In summary, we have demonstrated that electrolyte composition can serve as an effective degree to significantly improve the OER activity of electrocatalysts, such as Fe$_{1-y}$Ni$_y$S$_2$@Fe$_{1-x}$Ni$_x$OOH and SrBaNi$_2$Fe$_{12}$O$_{22}$, by rationally modulating its composition. Typically, when a co-electrolyte, HTAH aqueous solution, is introduced into the conventional KOH electrolyte, the strong adsorption of HTA$^+$ onto the surface of electrocatalysts renders modification of the zeta potential of the surface of shear for electrocatalysts. The preeminent OER activity of these two electrocatalysts was achieved at 0.02 M HTAH. Therefore, an intrinsic effect of HTAH on the OER activity of electrocatalysts can be attributed to an increase in the number of the O*···OH$^-$ pair in the diffusion double layer due to the electrostatic attraction between the adsorbed HTA$^+$ and OH$^-$ when the HTAH concentration is low, which significantly increases the OER rate. Moreover, the high stability of HTA$^+$ provides the high durability of OER electrolysis, and therefore promising applications in industrial electrolyzers. Concomitantly, we have prepared the new OER electrocatalysts including Fe$_{1-y}$Ni$_y$S$_2$@Fe$_{1-x}$Ni$_x$OOH microplatelets and SrBaNi$_2$Fe$_{12}$O$_{22}$, and more importantly, developed a new, effective strategy to enlarge the ECSA of electrocatalysts by the AB-assisted ECC. The reducing and capping effects of AB may play a key role in modulating the conversion rate of the surface electrocatalyst to oxyhydroxides, leading to the full growth of oxyhydroxide to

increase the accessible active site populations. Therefore, the proposed strategies of modulating electrolyte composition and ECSA of electrocatalysts prove to be effective in promoting highly active, resourceful, and durable OER electrocatalysis.

## Methods

**Synthesis of Fe$_{1-y}$Ni$_y$S$_2$@Fe$_{1-x}$Ni$_x$OOH/NF.** The synthesis of the Fe$_{1-y}$Ni$_y$S$_2$@Fe$_{1-x}$Ni$_x$OOH with a significantly enlarged ECSA involves the following four steps:

(i) Synthesis of the MIL-101 Fe precursor: typically, FeCl$_3$·6H$_2$O (0.675 g) and terephthalic acid (0.206 g) were fully dissolved in 15 mL of N,N-dimethylformamide by vigorous ultrasonic vibration. Next, the solution was transferred into a 45 mL Teflon-lined stainless steel autoclave. After the autoclave was sealed and heated to 110 °C and then maintained at this temperature for 20 h, it was cooled down to ambient temperature naturally. The brown product was isolated from the resulting reaction solution by centrifugation at 8000 r.p.m. and washed three times with 20 ml of ethanol at 60 °C to remove any unreacted starting materials. Finally, the product was dried at 60 °C in an oven for 8 h to obtain the red brown fine powder.

(ii) Synthesis of iron sulfides hexagonal microplatelets: In a typical synthesis, 1.4 mmol of sulfur powder (0.045 g) was dissolved in 2 mL of hydrazine hydrate (N$_2$H$_4$, 50–60%) at 60 °C. Afterward, the aqueous solution of sulfur was mixed with MIL-101 Fe (0.0903 g) and water (30 mL) under vigorous ultrasonic agitation in a 45 mL Teflon-lined stainless steel autoclave for 20 min, in order to form a homogeneous reaction mixture. Then, the autoclave was sealed and heated to 200 °C and then maintained for 6 h, followed by cooling down to ambient temperature naturally. Finally, the as-prepared product was separated from the reaction solution by centrifugation at 3000 r.p.m. for 3 min and decantation of the supernatant, and rinsed three times with 20 mL of a mixed solvent of ethanol and pyridine (1: 1, v/v), and three corresponding cycles of centrifugation and decanting of the supernatant. The resultant iron sulfides microplatelets were dried at 75 °C in an oven for 6 h for later use.

(iii) Synthesis of Ni-doped FeS$_2$ hexagonal microplatelets supported on NF (Fe$_{1-y}$Ni$_y$S$_2$/NF): Typically, a coating ink was prepared by ultrasonically dispersing 6 mg of iron sulfides microplatelets in 2 mL of anhydrous ethanol for 1 h at ambient temperature in a capped vial. Afterward, the ink was uniformly drop-casted onto a piece of cleaned NF with a size of 1.5 × 1.0 cm$^2$ to achieve a loading of 3 mg of iron sulfides per cm$^2$ of NF. After evaporating the solvent at 50 °C, the NF coated with iron sulfides microplatelets was loaded in the center of a horizontal tube furnace under vacuum of 10$^{-4}$ Pa and then heated to 350 °C at a ramping rate

of 5 °C min$^{-1}$, and kept at this temperature for 1.5 h under 1 atm of flowing high-purity argon gas. Thus, the integrated Fe$_{1-y}$Ni$_y$S$_2$/NF was obtained.

(iv) Synthesis of Fe$_{1-y}$Ni$_y$S$_2$@Fe$_{1-x}$Ni$_x$OOH/NF: The ECC of Fe$_{1-y}$Ni$_y$S$_2$/NF was carried out by CV cycling using a CHI 660D electrochemical analyzer and a conventional three-electrode system in an undivided clean polytetrafluoroethylene (PTFE) cell. The above-prepared Fe$_{1-y}$Ni$_y$S$_2$/NF, a Hg/HgO electrode (in 1 M KOH), and a carbon rod (to avoid catalyzing hydrolysis of AB) were used as the working electrode, reference electrode, and counter electrode, respectively. An Ar-purged aqueous solution containing 0.1 M KOH and 0.02 M AB was used as the working electrolyte. Continuous CV cycling the potential from −0.4 (lower limit) to 0.3 (upper limit) $V_{RHE}$ and back down to −0.4 $V_{RHE}$ (lower limit) for 20 cycles at a scan rate of 20 mV s$^{-1}$ was applied to the Fe$_{1-y}$Ni$_y$S$_2$/NF electrode. The electrochemically conditioned electrode, on which the active material has been converted to Fe$_{1-y}$Ni$_y$S$_2$@Fe$_{1-x}$Ni$_x$OOH, was subsequently washed with 10 mL of water at 45 °C three times, and then dried at 75 °C in an oven for 6 h. For comparison, ECC of Fe$_{1-y}$Ni$_y$S$_2$/NF was also conducted in an Ar-purged aqueous solution of 0.1 M KOH alone, but under otherwise conditions identical with those described above.

**Synthesis of SrBaNi$_2$Fe$_{12}$O$_{22}$ powders.** Typically, Ba(NO$_3$)$_2$ (261.4 mg, 1 mmol), Sr(NO$_3$)$_2$ (211.6 mg, 1 mmol), Ni(NO$_3$)$_2$·6H$_2$O (290.8 mg, 1 mmol), and Fe (NO$_3$)$_3$·9H$_2$O (404.0 mg, 1 mmol) in a 1:1:1:1 chemical stoichiometry were fully dissolved and mixed in 50 mL of H$_2$O under vigorous magnetic stirring until the solution became clear and transparent. Then, ethylenediaminetetraacetic acid disodium dihydrate (372.2 mg, 1 mmol) and citric acid monohydrate (420.2 mg, 2 mmol), which serve as the metal chelating agents, were added into the above solution of the mixed metal nitrates under vigorous stirring. After complete dis-solution of these two chelating agents, NH$_4$OH (25% NH$_3$) was added to adjust the pH of the solution to 6–8. Afterward, the solution was heated in an 80 °C water bath under vigorous stirring until it was converted into a gel state, followed by a drying process in an oven at 250 °C for 5 h. Next, the obtained black sponge-like bulk solid was heated to 1200 °C at a ramping rate of 10 °C min$^{-1}$, and then maintained at this temperature for 10 h in air in a horizontal tube furnace to obtain SrBaNi$_2$Fe$_{12}$O$_{22}$. The resulting product was naturally cooled to room temperature and then thoroughly ground to fine powders, using an agate mortar and a pestle.

**Electrochemical measurements.** To assess the OER performance, a three-electrode cell was used to measure the polarized CV curves, electrochemical impedance spectroscopy (EIS), ECSAs, and CP curves, where an active material, a calibrated Hg/HgO electrode (in 1 M KOH), and a Pt wire were used as the working electrode, reference electrode, and counter electrode, respectively. The Hg/HgO reference electrode was calibrated versus a RHE according to the procedure in a previous report of Burke et al.[20] The calibration was conducted in the high-purity hydrogen saturated electrolyte by bubbling H$_2$(g) over a freshly cleaned Pt gauze electrode. CVs were recorded at a sweep rate of 1 mV s$^{-1}$, and then the two potentials at which the current crossed zero were averaged as the thermodynamic potential for the hydrogen electrode reactions. For the hexaferrite, 3.0 mg of catalyst powders were prepared as homogeneous inks with 2.5 mL of a mixture of 45:50:5 of Milli-Q water, 2-propanol, and 5 wt % Nafion solution by ultrasonication for at least 30 min. Then, 17 μL of the inks was drop-casted onto a glassy carbon disc (GCD) electrode (Pine Research Instrumentation, 5 mm in diameter) and dried naturally, yielding a catalyst loading of 0.25 mg cm$^{-2}$. For comparison, RuO$_2$ nanoparticles, which were prepared according to a recent report[52], were homogeneously dispersed into the same solvent as above, and pipetted and spread on NF to obtain the same catalyst loading as that of the Fe$_{1-y}$Ni$_y$S$_2$@Fe$_{1-x}$Ni$_x$OOH/NF. Each active material with a loading of 3 mg cm$^{-2}$ on NF that was tailored into an exposure area of 1 × 1 cm$^2$ was directly used as working electrode. All the electrochemical mea-surements were carried out in clean PTFE electrochemical test cells obtained by rinsing with 2 M H$_2$SO$_4$ at 50 °C overnight, and then hot water at 80 °C five times to eliminate the impacts of any impurities. All the electrolytes are Fe-free electrolytes purified by fresh Ni(OH)$_2$ precipitates according to the method by Trotochaud et al.[17] The electrolytes are ultrahigh purity O$_2$-saturated aqueous solutions of 1 M OH$^-$ ions (1 M KOH alone or total 1 M OH$^-$ concentration comprised of KOH and HTAH with varied fractions), which were obtained by bubbling O$_2$ for 20 min prior to each experiment and continuously bubbling during the data collection. All the measurements were performed using a CHI 660D electrochemical analyzer (CHI Instruments Inc., Shanghai). All the measured potentials were referenced to the RHE, according to the Nernst equation $V_{RHE} = V_{Ag/AgCl} + E_{Ag/AgCl} + 0.059$ pH, where $V_{RHE}$ and $V_{Ag/AgCl}$ are the applied potentials against the RHE and Ag/AgCl reference electrode, respectively, and $E_{Ag/AgCl}$ is the reference electrode potential versus the standard hydrogen electrode. To reach the reliable OER activity, all the working electrodes were first subjected to continuous potential cycling between 1.0 and 1.8 $V_{RHE}$ at a scan rate of 20 mV s$^{-1}$ in an oxygen-saturated electrolyte until stabilized voltammograms were obtained. Then, all the measured currents were normalized by the exposed geometric surface area of the electrode. EIS measurements were performed by sampling 100 points in the frequency range from 100 kHz to 0.01 Hz, with an AC perturbation of 5 mV at ambient temperature. The complex nonlinear least square fitting of all the EIS spectra was conducted with the Zview 3.1 software package. All polarization curves were iR-corrected according to the equation $E = E_m - iR$ (where $E$ is the corrected potential, $E_m$ is the measured potential, and $R$ is the resistance of the solution).

TOF, which is defined as the number of O$_2$ molecules evolved from per metal active site per second based on the total number of Fe and Ni atoms of the Fe$_{1-x}$Ni$_x$OOH shell[20,37], is utilized to compare intrinsic activities of electrode electrocatalysts in different electrolytes. The composition and weight ratio of the Fe$_{1-x}$Ni$_x$OOH shell of Fe$_{1-y}$Ni$_y$S$_2$@Fe$_{1-x}$Ni$_x$OOH electrocatalyst were determined by EDX analysis in combination with XPS measurement, which give its mass of 0.271 mg cm$^{-2}$ (i.e., $3.027 \times 10^{-6}$ mol cm$^{-2}$). The amount of the evolved O$_2$ and FE of the OER, operated at an iR-corrected $\eta$ of 300 mV, were determined by GC (Shimadzu, GC-8A) with a thermal conductivity detector and N$_2$ as the carrier gas.

The zeta (ζ) potentials of the Fe$_{1-y}$Ni$_y$S$_2$@Fe$_{1-x}$Ni$_x$OOH/NF electrode in various electrolytes containing a different concentration of HTAH were determined using a streaming potential analyzer (DelsaNano C/Solid Surface, Beckman).

## Data availability

The authors declare that all data supporting the findings of this study are available within the article and Supplementary Information files, and also from the corresponding author upon reasonable request.

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

## Acknowledgements

This work was financially supported by the Six Talent Peaks Project in Jiangsu Province (Grant No. JNHB-043), the Research Fund of State Key Laboratory of Materials-Oriented Chemical Engineering (Grant No. ZK201713), the Key University Science Research Project of Jiangsu Province (Grant No. 16KJA150004), and the National Natural Science Foundation of China (Grant No. 21371097).

## Author contributions

Y.T. developed the idea, analyzed the data, and wrote the manuscript. Y.G. performed the experiments and characterized the materials, C.W. and S.Y. assisted Y.G. with the materials synthesis, characterization, and electrochemical measurements. All of the authors discussed the results and conducted data analysis.

## Competing interests

The authors declare no competing interests.
