## [Peer Review File · Communications Chemistry]

Reviewers' comments:

Reviewer #1 (Remarks to the Author):

In this manuscript, Gao et al. have demonstrated that electrolyte composition can serve as an effective degree to significantly improve the OER activity of ECs, such as Fe_{1-y}Ni_yS₂@Fe_{1-x}Ni_xOOH and SrBaNi₂Fe₁₂O₂₂, by rationally modulating its composition. Moreover, the high stability of HTA⁺ provides the high durability of OER electrolysis and therefore promising applications in industrial electrolyzers. After careful evaluation of the paper, I recommend publication subject to a minor revision in the following aspects.

1. The authors wrote: "A series of high-performance OER ECs, such as Ni-Fe oxyhydroxides, cobalt-based nanomaterials, and perovskite oxides, have received intensive interest due to their remarkable activities, excellent stability, and low cost, which may expedite their widespread commercialization". Based on the composition of the study material, that is, Fe_{1-y}Ni_yS₂@Fe_{1-x}Ni_xOOH (MOF-derived sulfides@oxyhydroxides), it is also important to mention the recent and promising materials: MOF-derived sulfides [doi: 10.1016/j.electacta.2020.135636 and 10.1016/j.ijhydene.2019.11.156] and NiV oxide/hydroxide [doi: 10.1039/C9TA10857B].

2. Please provide the magnification values in the SEM images for Figures 1a, b and c.

3. The authors wrote: "The Nyquist plots (symbols) of EIS centered at 1.57 V vs. reversible hydrogen electrode (VRHE) according to modeling with the corresponding electric equivalent circuit (EEC) reveal the presence of two overlapped semicircles at high frequencies (HF) and low frequencies (LF), respectively". A detailed assignment of high and low frequency charge transfer processes is missing.

4. The diffusion layer was mentioned in the body of the manuscript, however, no diffusional element was incorporated in the equivalent circuit. Please make a further explanation about the equivalent circuit.

5. The authors wrote: "In particular, the Fe_{1-y}Ni_yS₂@Fe_{1-x}Ni_xOOH/NF exhibits the lowest Tafel slopes with a value of 35.5 mV dec⁻¹ in the low j region and 36.9 mV dec⁻¹ in the large j region in KOH + HTAH, which is slightly higher than that (33.6 mV dec⁻¹) in the lower j region but much lower than the value (70.4 mV dec⁻¹) of the larger j in KOH, respectively". The value of the Tafel slope in the lower j region in KOH written in the text is different from that written in Figure 3c.

6. The inset in panel (b) of Supplementary Fig. 7 is very difficult to see. Please insert a better quality figure.

Reviewer #2 (Remarks to the Author):

In this manuscript, the authors developed an effective method to improve the OER activities of Fe_{1-y}Ni_yS₂@Fe_{1-x}Ni_xOOH via the introduction of hexadecyltrimethylammonium hydroxide (HTAH) into the KOH electrolyte. The OER performance comparisons of SrBaNi₂Fe₁₂O₂₂ powders were also tested as a verification. The manuscript is well written and organized, and experimental data are relatively sufficient and convincing. Hence, the present work is suggested to be published in Communications Chemistry after Major revision. Detailed comments are given as follows:

1. Authors are advised to cut down on unnecessary abbreviations, such as "ECs" for "electrocatalysts", and "MPs" for "Microplates".

2. Figures 1f-g are not clear enough, authors are suggested to supply clearer figures.

3. XRD patterns are suggested to change to a clearer version since some peaks and numbers are stacked in Figure 2b and 6b.

4. For the OER performance comparison, authors are suggested to compare the overpotential of electrocatalysts under the current density of 10 mA cm⁻², which is more commonly used in the literatures.

5. More advanced materials have been developed and should be discussed in Introduction. Some suggested references are following: Matter, 2, 526-553, 2020, Journal of Materials Chemistry A, 7, 9386-9405, 2019, Small Methods, 2, 1800001, 2018, ChemElectroChem, 5, 1424-1434, 2018, Journal of Materials Chemistry A, 8, 10604-10624, 2020, Advanced Functional Materials, 1910768, 2020, Small Methods, 4, 1900575, 2020

6. Since the KOH concentrations of the electrolytes in this work are different, the authors are suggested to calibrate the reference electrode to RHE in all the electrolytes.

Reviewer #3 (Remarks to the Author):

Comments :

In this manuscript, the authors demonstrated an interesting strategy that introducing hexadecyltrimethylammonium hydroxide (HTAH) into the alkaline electrolyte can dramatically enhance the OER performance. It's explained that THA⁺ cations strongly adsorbed on catalyst surface could modify the surface and increase the absolute number of OH⁻ ions near the EC surface. Based on these observations and hypothesis, it's claimed that tuning the electrolyte composition in combination with engineering ECs would open a new and efficient way to promote the performance of OER electrolysis. This work reflects the complicated surface interactions on OER catalysts. Results are interesting and shed light on the principles for designing of highly active TM chalcogenide catalysts. However, the conclusion is not clear, and several points should be clarified prior to the publication.

1) The main point of this work is to probe the unique role of hexadecyltrimethylammonium hydroxide in promoting the OER activities, which is expected to be a universal strategy. I don't understand why the authors use a very complicated catalyst system such as Fe_{1-y}Ni_yS₂@Fe_{1-x}Ni_xOOH for demonstration. Nevertheless, many studies have shown that Fe_{1-y}Ni_yS₂ is indeed not stable as OER catalyst, which inevitably converts to hydroxides after cycling.

2) As shown in Figure 3a and supplementary Figure 15, it's noted that adding THA⁺ cations can effectively enlarge the peak areas of the Ni^{2+/3+} redox. Then, is it possible that the THA⁺ cations in the electrolyte exfoliate the layered structure of NiFe hydroxides resulting in enlarged electrochemical surface area for OER reaction? If so, the final conclusion could be misleading.

3) Authors claimed that the adsorption of THA⁺ cations on catalyst surface can increase the absolute number of OH⁻ ions which leads to enhance OER performance. However, this claim is somehow questionable. The OER is progressed at high potentials during which adsorbed OH⁻ ions are quickly converted to oxygen molecules, with the active metal sites being quickly refilled by OH⁻ ions from the electrolyte. However, adsorption of THA⁺ cations may change the hydrophobic properties of the catalyst surface and indeed slow down this refilling step.

4) Since the electrode is charged at high potentials during the OER, the interactions between cations and catalyst surface are highly dynamic and may be significantly different from the uncharged state. In this case, the zeta potentials measured for catalyst powders in different electrolyte may not be able to provide vital information for such interactions.

5) Finally, it's essential to measure the pH of mixed electrolyte properly. Whether adding small amount of hexadecyltrimethylammonium hydroxide can change the pH of the alkaline electrolyte? It's better to provide the pH of the mixed solutions in the manuscript or the SI.

We have carefully and thoroughly revised our manuscript according to the reviewers' comments and answered the questions raised by them. Herein, we provide the point-by-point explanations and answers in the sequence of the comments. In the revised manuscript, we provide new explanations and experimental data to consolidate our conclusions and further improve the quality of our manuscript. All the changes that we made have been highlighted with yellow background in the revised manuscript. In particular, we would like to express our sincere thanks to all the reviewers for their very careful, professional, and high-level comments that are very helpful for us to improve the quality of our manuscript.

For Reviewer 1:

1. The authors wrote: "A series of high-performance OER ECs, such as Ni-Fe oxyhydroxides, cobalt-based nanomaterials, and perovskite oxides, have received intensive interest due to their remarkable activities, excellent stability, and low cost, which may expedite their widespread commercialization". Based on the composition of the study material, that is, $\text{Fe}_{1-y}\text{Ni}_y\text{S}_2@\text{Fe}_{1-x}\text{Ni}_x\text{OOH}$ (MOF-derived sulfides@oxyhydroxides), it is also important to mention the recent and promising materials: MOF-derived sulfides [doi: 10.1016/j.electacta.2020.135636 and 10.1016/j.ijhydene.2019.11.156] and NiV oxide/hydroxide [doi: 10.1039/C9TA10857B].

Response: The suggested MOF-derived sulfides and NiV oxide/hydroxide have been addressed in the text (please see line 10 and 11 in the second paragraph on page 1). The related references are cited in the text (please see ref. 18, 21, and 22 on page 9).

2. Please provide the magnification values in the SEM images for Figures 1a, b and c.

Response: The magnification values for the SEM images in Figures 1a, b, and c are provided in the figure caption of Fig. 1 (please see the last line of the figure caption of Fig. 1).

3. The authors wrote: "The Nyquist plots (symbols) of EIS centered at 1.57 V vs. reversible hydrogen electrode (VRHE) according to modeling with the corresponding electric equivalent circuit (EEC) reveal the presence of two overlapped semicircles at high frequencies (HF) and low frequencies (LF), respectively". A detailed assignment of high and low frequency charge transfer processes is missing.

Response: The detailed assignment of high and low frequency charge transfer processes had been provided in the original submission (please see the first paragraph on page 12 in the Supplementary Information). The reviewer probably overlooked it. For clarity, the supplementary specification is added in line 5 and 6 in the second paragraph on page 4.

4. The diffusion layer was mentioned in the body of the manuscript, however, no diffusional element was incorporated in the equivalent circuit. Please make a further explanation about the equivalent circuit.

Response: It should be pointed out that the EIS gives the information of electrode response rather than the configuration of electrolyte solution. The effects of the diffusion layer can be detected by the changes of the R_{ct} value of electrocatalysts based on the fitted equivalent circuit.

5. The authors wrote: "In particular, the $\text{Fe}_{1-y}\text{Ni}_y\text{S}_2@\text{Fe}_{1-x}\text{Ni}_x\text{OOH}/\text{NF}$ exhibits the lowest Tafel slopes with a value of 35.5 mV dec^{-1} in the low j region and 36.9 mV dec^{-1} in the large j region in $\text{KOH} + \text{HTAH}$, which is slightly higher than that (33.6 mV dec^{-1}) in the lower j region but much lower than the value (70.4 mV dec^{-1}) of the larger j in KOH , respectively". The value of the Tafel slope in the lower j region in KOH written in the text is different from that written in Figure 3c.

Response: Indeed, there is a typing error. Please forgive us for the carelessness. In the text of the revised

version, the value has been changed to 32.9 (please see line 13 in the second paragraph on page 4).

6. The inset in panel (b) of Supplementary Fig. 7 is very difficult to see. Please insert a better quality figure.

Response: The quality of the inset in panel (b) of Supplementary Fig. 7 has been improved. Please see the new inset in Supplementary Fig. 7b.

For Reviewer 2:

1. Authors are advised to cut down on unnecessary abbreviations, such as “ECs” for “electrocatalysts”, and “MPs” for “Microplates”.

Response: All the unnecessary abbreviations, such as “ECs” for “electrocatalysts”, “MPs” for “microplatelet”, “NKs” for nanoflakes, and “NSs” for nanosheets have been replaced by the corresponding full name.

2. Figures 1f-g are not clear enough, authors are suggested to supply clearer figures.

Response: In light of the reviewer’s suggestions, the clearer HRTEM images are provided in Figures 1f and g.

3. XRD patterns are suggested to change to a clearer version since some peaks and numbers are stacked in Figure 2b and 6b.

Response: The XRD patterns with clearer markers for the indices of crystallographic planes are provided in Figure 2b and 6b.

4. For the OER performance comparison, authors are suggested to compare the overpotential of electrocatalysts under the current density of 10 mA cm^{-2} , which is more commonly used in the literatures.

Response: On the one hand, the low current densities (for example, at 10 mA cm^{-2}) for water oxidation are interfered with the oxidation wave of Ni^{2+} ions. On the other hand, in order to carry the practical importance based on the industrial water electrolyzers operating at large current densities (for example, $200\text{--}400 \text{ mA cm}^{-2}$ in alkaline media and at $600\text{--}2000 \text{ mA cm}^{-2}$ in acidic media, please see Carmo et al. *Int. J. Hydrogen Energy* **38**, 4901 (2013)), the assessment and comparison of overpotentials at larger densities (typically at 100 mA cm^{-2}) are preferable.

5. More advanced materials have been developed and should be discussed in Introduction. Some suggested references are following: Matter, 2, 526-553, 2020, Journal of Materials Chemistry A, 7, 9386-9405, 2019, Small Methods, 2, 1800001, 2018, ChemElectroChem, 5, 1424-1434, 2018, Journal of Materials Chemistry A, 8, 10604-10624, 2020, Advanced Functional Materials, 1910768, 2020, Small Methods, 4, 1900575, 2020.

Response: The progress of the suggested advanced materials that have been recently developed has been discussed in the Introduction section. Please see line 6–9 in the second paragraph on page 1 in the text. The related references are cited in the revised manuscript (please see ref. 10–13 on page 9). In addition, some other reports are focused on hydrogen evolution reaction and electrochemical energy-storage (supercapacitors) (*Matter* 2, 526–553, (2020)), carbon dioxide reduction (*Adv. Funct. Mater.* 30, 1910768, (2020)), and oxygen reduction reaction (*Small Methods* 4, 1900575, 2020), which are irrelevant to our studies.

6. Since the KOH concentrations of the electrolytes in this work are different, the authors are suggested to calibrate the reference electrode to RHE in all the electrolytes.

Response: Actually, as described in the original submission, the reference electrode in the different electrolytes has been calibrated to RHE before the electrochemical measurements in a different electrolyte

(please see line 4 on page 8). The reviewer probably overlooked it. For clarity and rigidity, the detailed calibration procedure is provided in the text. Please see line 5–7 on page 8.

For Reviewer 3:

1. The main point of this work is to probe the unique role of hexadecyltrimethylammonium hydroxide in promoting the OER activities, which is expected to be a universal strategy. I don't understand why the authors use a very complicated catalyst system such as $\text{Fe}_{1-y}\text{Ni}_y\text{S}_2@\text{Fe}_{1-x}\text{Ni}_x\text{OOH}$ for demonstration. Nevertheless, many studies have shown that $\text{Fe}_{1-y}\text{Ni}_y\text{S}_2$ is indeed not stable as OER catalyst, which inevitably converts to hydroxides after cycling.

Response: As clarified in numerous previous reports, the true active components of various electrocatalysts for catalyzing water oxidation are metallic oxyhydroxides that are in situ grown on metallic oxides, chalcogenides, phosphides, and so on during the OER. Meanwhile, the metallic oxyhydroxides generally act as a protective layer to prevent the inner bulk components from further conversion to oxyhydroxides or dissolution into electrolytes. Therefore, these electrocatalysts with a core@shell architecture are stable enough for water oxidation, as stated in our studies. The good stability of bimetallic sulfides for the OER can be found in refs 11, 18, 21, 22, and so on.

2. As shown in Figure 3a and supplementary Figure 15, it's noted that adding THA^+ cations can effectively enlarge the peak areas of the $\text{Ni}^{2+}/\text{Ni}^{3+}$ redox. Then, is it possible that the THA^+ cations in the electrolyte exfoliate the layered structure of NiFe hydroxides resulting in enlarged electrochemical surface area for OER reaction? If so, the final conclusion could be misleading.

Response: Thank the reviewer very much for this question. The ECSAs measurements reveal that The ECSAs for each active material are very similar in both electrolytes, as extensively proofed in the text. Obviously, the enlarged peak areas of the $\text{Ni}^{2+}/\text{Ni}^{3+}$ redox couple originates from the accelerated reaction rate, further indicating the increased concentration of OH^- ions near the electrocatalyst surface based on the reaction equation: $\text{Ni}^{2+} + \text{H}_2\text{O} + \text{OH}^- = \text{NiOOH} + \text{e}^- + 2\text{H}^+$. Please see the explanations in the last line on page 5 and the first three lines on page 6.

In addition, no surfactants have shown the so powerful function of exfoliation of the layered hydroxides in the literature (for example, see ref 36).

3. Authors claimed that the adsorption of THA^+ cations on catalyst surface can increase the absolute number of OH^- ions which leads to enhance OER performance. However, this claim is somehow questionable. The OER is progressed at high potentials during which adsorbed OH^- ions are quickly converted to oxygen molecules, with the active metal sites being quickly refilled by OH^- ions from the electrolyte. However, adsorption of THA^+ cations may change the hydrophobic properties of the catalyst surface and indeed slow down this refilling step.

Response: We do not believe that the reviewer's assumption is the fact. First, we used a low concentration of HTAH, which means that the effect of HTAH on the hydrophobicity of electrode surface is minor. More importantly, the refilling process for OH^- ions must be accelerated due to the electrostatic attraction in the presence of the adsorbed HTA^+ . In contrast, this refilling process is slow if no HTA^+ cations are present on the electrode surface, because the diffusion of OH^- ions towards electrode surface is only determined by their concentration gradient.

4. Since the electrode is charged at high potentials during the OER, the interactions between cations and catalyst surface are highly dynamic and may be significantly different from the uncharged state. In this case, the zeta potentials measured for catalyst powders in different electrolyte may not be able to provide vital information for such interactions.

Response: Although the electrode is charged at high potentials, the change of electrode potential does not alter the adsorption properties of ions and surfactants on electrode surface due to the strong adsorption energies of these species and uniform nature of catalytic electrode (please see Yang et al. *Angew. Chem. Int. Ed.* **56**, 8652–8656 (2017) and Garcia et al. *Angew. Chem. Int. Ed.* **58**, 12999–13003 (2019)). For example, the energy change is only 1 eV when potential is increased from 1 to 2 V_{RHE}, however, the adsorption energies of ions and surfactants on solid surface are usually several eV.

5. Finally, it's essential to measure the pH of mixed electrolyte properly. Whether adding small amount of hexadecyltrimethylammonium hydroxide can change the pH of the alkaline electrolyte? It's better to provide the pH of the mixed solutions in the manuscript or the SI.

Response: the pH values of all the mixed electrolytes are given in line 13–16 in the second paragraph on page 5 in the text. The measurement is given in the last two lines in the first paragraph on page 2. It can be seen that the pH values of the electrolytes containing a different HTAH concentration do not differ much.

REVIEWERS' COMMENTS:

Reviewer #1 (Remarks to the Author):

The authors have addressed all of the questions raised by reviewers, thus I would recommend publication of this work in its present form.

Reviewer #2 (Remarks to the Author):

It can be published.

Reviewer #3 (Remarks to the Author):

Authors have carefully answered my questions point by point and revised the manuscript accordingly. The electrocatalysts were systematically characterized and the concept is novel and interesting, which should arouse readers' interest in the field. Although I am still not fully convinced for the proposed mechanism, for the current stage, the manuscript is worth for publication.